# circRNA: A New Biomarker and Therapeutic Target for Esophageal Cancer

**DOI:** 10.3390/biomedicines10071643

**Published:** 2022-07-08

**Authors:** Katsutoshi Shoda, Yuki Kuwano, Daisuke Ichikawa, Kiyoshi Masuda

**Affiliations:** 1First Department of Surgery, Faculty of Medicine, University of Yamanashi, Yamanashi 409-3898, Japan; kshoda@yamanashi.ac.jp (K.S.); dichikawa@yamanashi.ac.jp (D.I.); 2Department of Medical Genetics, Institute of Biomedical Sciences, Tokushima University Graduate School, Tokushima 770-8503, Japan; kuwanoy@tokushima-u.ac.jp; 3Kawasaki Medical School, Okayama 701-0192, Japan

**Keywords:** circRNA, esophageal cancer, biomarker, therapeutic target

## Abstract

Circular RNAs (circRNAs) comprise a large class of endogenous non-coding RNA with covalently closed loops and have independent functions as linear transcripts transcribed from identical genes. circRNAs are generated by a “back-splicing” process regulated by regulatory elements in cis and associating proteins in trans. Many studies have shown that circRNAs play important roles in multiple processes, including splicing, transcription, chromatin modification, miRNA sponges, and protein decoys. circRNAs are highly stable because of their closed ring structure, which prevents them from degradation by exonucleases, and are more abundant in terminally differentiated cells, such as brains. Recently, it was demonstrated that numerous circRNAs are differentially expressed in cancer cells, and their dysfunction is involved in tumorigenesis and metastasis. However, the crucial functions of these circRNAs and the dysregulation of circRNAs in cancer are still unknown. In this review, we summarize the recent reports on the biogenesis and biology of circRNAs and then catalog the advances in using circRNAs as biomarkers and therapeutic targets for cancer therapy, particularly esophageal cancer.

## 1. Introduction

Covalently closed circular RNAs (circRNAs) are a large class of non-coding RNAs that are generated by a process called back-splicing. In 1976, the first circular RNA molecule (viroid) was discovered as a pathogen by Sanger sequencing [1]. After a few years, Hsu et al. [2] reported the presence of the circular form of RNA in the cytoplasmic extraction of several eukaryotic cells using electron microscopy. Since then, a variety of circular RNAs have been reported. In 1991, Nigro et al. discovered some abnormally spliced transcripts called “scrambled exons”, which originated from the candidate tumor suppressor gene *deleted in colorectal cancer* (*DCC*) and were produced by non-canonical splicing [3]. A year later, similar transcripts from the human *ETS proto-oncogene 1, transcription factor* (*ETS-1*) gene were reported [4]. In 1993, Cocquerelle et al. identified that these mis-splicing products from the *ETS-1* and *DCC* genes are circular RNA molecules containing only normally spliced exons and are localized in the cytoplasm [5]. The second report in 1993 demonstrated that the *Sry* RNA molecules in cDNA and 5′RACE clones isolated from mouse testes have a circular structure [6]. This circular RNA represents the most abundant transcript in the cytoplasm and testis, indicating that it might be functional. Several other studies have documented that exon circularization can be induced in nuclear extracts in vitro [7,8]. In recent years, the advancement of high-throughput RNA sequencing (RNA-seq) followed by bioinformatic algorithms specialized for circRNAs has identified thousands of circRNAs in metazoans, including *Drosophila*, nematodes, mice, and humans, and have shown their tissue- and developmental-specific expression pattern [9,10,11,12,13,14].

Most circRNAs are generated by back-splicing, in which a downstream splice donor site is covalently linked to an upstream splice-acceptor site, and are abundant in the cytoplasm. Bioinformatic algorithms that identified circRNAs from long-read RNA-seq data revealed that more than half of the circRNAs span <5 complete exons and consist of only protein-coding exons, especially in 5′-untranslated regions (UTRs) [15]. In contrast to exon-derived circRNAs, failure in intron lariat debranching or internal intron retention during canonical splicing may produce circular intronic RNAs (ciRNAs) and exon–intron circular RNAs (EIciRNAs), respectively. These classes of circRNAs are abundant in the nucleus and act as stimulators of the transcription of their parental genes in cis [16,17].

In general, circRNAs are highly stable because of the covalently closed ring structure that protects them from degradation by exonucleases [10] and are more abundant in terminally differentiated cells, such as brains [11]. In contrast, the expression levels of circRNAs in highly proliferating cells, such as cancer cells, are often low, possibly because of the thinning out by cell proliferation [18].

Recent evidence has shown that circRNAs participate in many pathological processes, such as diabetes mellitus, Alzheimer’s disease, atherosclerosis, chronic inflammatory diseases, and cancer [19,20,21,22,23]. This review summarizes the recent reports on the biogenesis and biology of circRNAs. We also cataloged the advances in using circRNAs as biomarkers and therapeutic targets for esophageal cancer.

## 2. Biogenesis of circRNAs

Many studies have shown that circRNAs are expressed at much lower levels than their associated linear transcripts, and their expression levels do not simply correlate with their linear isoform expression, indicating a potential layer of unknown regulation [24]. Some studies have revealed that circRNA production strongly depends on the presence of canonical splice sites in bracketing exons and spliceosome assembly, indicating that circRNA biogenesis competes with the linear splicing of flanking exons during the canonical splicing machinery [25,26]. However, two reports have shown that depleting the splicing factors or components of U2 snRNPs increases the ratio of circRNAs to linear RNAs [27,28], indicating that the suppression or slowing of canonical pre-mRNA splicing machinery changes the steady-state production of linear transcripts to circular RNAs.

circRNA formation depends on two mechanisms. The first is base pairing between inverted repeat elements located in both the upstream and downstream introns [29]. In humans, 88% of these inverse-repeat elements are Alu repeats [10,30,31]. Double-stranded RNA (dsRNA)-specific adenosine deaminase (ADAR), which mediates adenosine to inosine editing in endogenous dsRNA, and DexH-Box helicase 9 (DHX9), which catalyzes the ATP-dependent unwinding of double-stranded RNA complexes, suppress the production of circRNAs that depend on base pairing between inverted repeats to prevent the looping of intron sequences [31,32]. In contrast, nuclear factor 90 (NF90) and 110 (NF110), both of which contain a nucleic acid-binding motif (double-stranded RNA-binding motif, dsRBM), promote the biogenesis of circRNAs by stabilizing the base pairing between inversed repeats. The second mechanism involves the dimerization of RNA-binding proteins (RBPs). Quaking (QKI), which belongs to the STAR family of KH domain-containing RBPs, is associated with regions containing recognition elements within a single RNA, resulting in the stimulation of circRNA production [33]. FUS, also known as hnRNPP2, binds to intron regions proximal to the splice junctions involved in circRNA formation and affects the back-splicing reaction [34]. The other RBP of the heterogeneous nuclear ribonucleoprotein family, hnRNPL, binds preferentially to CA-repeat or CA-enriched RNA motifs in introns, resulting in enhanced circulation [35].

After processing in the nucleus, most exon-derived circRNAs are exported into the cytoplasm. RNAi screening assays have revealed that the DEAD box family proteins DDX39A and DDX39B transport circRNAs from the nucleus to the cytoplasm in a size-dependent manner [36]. 

## 3. Biological Roles of circRNAs

circRNAs can stably exist in various subcellular fractions, such as the nucleus, cytoplasm, ribosome, cytosol, and exosome, because their loop structures are not affected by the 3′-to-5′ RNA exonuclease RNase R. In general, circRNAs have independent functions from linear transcripts transcribed from the same host genes because of their longer half-lives. Increasing evidence has shown that circRNAs can associate with other regulatory factors, such as microRNAs (miRNAs) and RBPs, and participate in multiple processes, including splicing, transcription, chromatin modification, miRNA sponges, and protein decoys. Here, we summarize the biological functions of circRNAs, particularly with regard to cancer development.

### 3.1. miRNA Sponges

MiRNAs are post-transcriptional regulators that directly bind to mRNAs and inhibit translation or lead to mRNA degradation [37]. In recent years, increasing evidence has suggested that circRNA–miRNA–mRNA regulatory networks are important for exploring the pathogenesis and therapeutic strategies of cancer. One of the most widely investigated functions of circRNAs is to regulate posttranscriptional gene expression by acting as miRNA sponges. These circRNAs localize in the cytoplasm and contain multiple miRNA-binding sequences [10]. CiRS-7 (also termed cerebellar degeneration-related protein 1 antisense RNA (CDR1as)) was the first identified circRNA to harbor more than 70 binding sites for miR-7. MiR-7 is a tumor-suppressive miRNA involved in several pathophysiological pathways in hepatocellular carcinoma, breast cancer, and gastric cancer [38]. CiRS-7 prevents the degradation of miR-7-targeted mRNAs by functioning as a competing endogenous RNA [39]. 

To date, dozens of cancer-related circRNAs have been found to participate in the occupation of miRNA response elements. Hsa_circRNA_0088036 expression is upregulated in bladder cancer tissues and promotes cancer development by competing with miR-140-3p, resulting in the induction of FOXQ1 [40]. Fang et al. reported that circRNA phenylalanyl-tRNA synthetase subunit alpha (circFARSA) interacts with miR-330 and increases cell proliferation and invasion in bladder cancer tissues [41]. Hsa_circ_0000567 can act as a sponge for miR-421, which increases cell migration and invasion by directly binding to the 3′-UTR of *TMEM100* mRNA in lung adenocarcinoma [42]. Downregulation of circ_0000567 accelerates the development of lung adenocarcinoma via the hsa_circ_0000567/miR-421/TMEM100 axis. CircSYPL1 expression is upregulated in patients with hepatocellular carcinoma; circSYPL1 sponges miR-506-3p to elevate EZH2 expression and induce tumorigenesis in hepatocellular carcinoma cells [43]. 

### 3.2. Epigenetic Regulation

Several studies have shown that circRNAs are involved in epigenetic regulation, including histone and chromatin modifications. circIMMP2L mediates the malignancy of esophageal squamous cell carcinoma (ESCC) by promoting the nuclear retention of CtBP1 [44]. In the nucleus, circIMMP2L mediates the interaction between CtBP1 and HDAC1 and induces the deacetylation of histone H3 in the promoter regions of *E-cadherin* and *p21*. CircMRPS35 recruits histone acetyltransferase KAT7 to the *FOXO1* and *FOXO3a* promoter regions, leading to the acetylation of H4K5, which facilitates the activation of *FOXO1/3a* transcription [45]. FOXO1/3a affects the expression of downstream genes such as *p21*, *Twist1*, and *E-cadherin*, resulting in the suppression of cell proliferation and invasion.

### 3.3. Transcription and Alternative Splicing

Nuclear circRNAs can regulate gene expression by affecting transcription and alternative splicing, whereas cytoplasmic circRNAs function through interactions with miRNAs or proteins. EIciRNAs, mainly located in the nucleus, interact with U1 snRNP and promote the RNA Pol II-mediated transcription of their parental genes [17]. CircITGA7 downregulates colorectal cancer cell proliferation via facilitating the transcription of its host gene *ITGA7* [46]. Ci-ankrd52, a circular intronic RNA generated from the *ANKRD52* gene, is recruited to its transcription sites to promote RNA Pol II transcription [16]. 

The effects of circRNAs on pre-mRNA splicing have also been well characterized. CircRPAP2, a circRNA derived from the host gene *RPAP2*, is expressed in breast cancer tissues [47] and inhibits the proliferation and migration of breast cancer by competing with the association between the splicing factors SRSF1 and *PTK2* pre-mRNA to modify the alternative splicing pattern of *PTK2* mRNA. CircURI1 directly associates with hnRNPM to modulate the alternative splicing of a subset of genes involved in cell migration, resulting in the suppression of gastric cancer metastasis [48].

### 3.4. Protein Decoys

Several circRNAs harbor binding sites for RBPs and act as decoys to suppress protein function. For example, circMbl is derived from the splicing factor *muscleblind* (*mbl*) locus, which harbors multiple binding sites for the MBL protein [25]. CircMbl regulates MBL function by preventing its binding to its target RNAs. CircPABPN1 binds to Hu antigen R (HuR, also known as ELAV-like RNA-binding protein (ELAVL1)) to prevent the association of HuR with *PABPN1* mRNA, resulting in the suppression of the effects of HuR on *PABPN1* translation [49]. Circular antisense non-coding RNA in the *INK4* locus (circANRIL) interacts with pescadillo homolog 1 (PES1) to disrupt pre-rRNA processing mediated by exonuclease in vascular smooth muscle cells [21]. Thus, circANRIL can control ribosome biogenesis to induce nucleolar stress and p53 activation, resulting in the stimulation of apoptosis, which is related to atherosclerosis pathology. Circ-transportin 3 (TNPO3) acts as a protein decoy for insulin-like growth factor 2 binding protein 3 (IGF2BP3) to interfere with the MYC/SNAIL axis, thereby decreasing the proliferation and metastasis of gastric cancer [50]. Chen et al. reported that circ_0000079 associates with Fragile X-Related 1 (FXR1) to interrupt the formation of the FXR1/protein kinase C, iota (PRKCI) complex, which mediates the inhibition of cell invasion and drug resistance in non-small cell lung cancer [51].

### 3.5. Translation into Peptides/Proteins

Although circRNAs have been categorized as non-coding RNAs, some circRNAs that harbor the open reading frame driven by internal ribosome entry sites are potentially translated into peptides or proteins [52,53]. Circ-ZNF609 exhibits internal ribosome entry site-dependent translation and functions in myoblast proliferation [54]. Zhang et al. reported that the circ-SNF2 histone linker PHD RING helicase (SHPRH) encodes a novel protein, SHPRH-146aa [55]. This truncated protein prevents the full-length SHPRH protein from degradation by the ubiquitin-proteasome and functions as a tumor suppressor in glioblastoma. CircMAPK14-175aa, a peptide of 175 amino acids encoded by the *circMAPK14* gene, reduces the nuclear translocation of MAPK14 by competitively binding to MKK6 and blocking the progression of colorectal cancer [56]. Recent studies have shown that N6-methladenosine (m6A) RNA methylation can drive translation initiation in circRNAs [57]. circRNAs containing m6A are recognized by YTHDF3 and the translation initiation factor eIF4G2. Driven by m6A modification, the m6A reader protein IGF2BP1 promotes circMAP3K4 translation into circMAP3K4-455aa in hepatocellular carcinoma [58]. The expression of circMAP3K4-455aa prevents cisplatin-induced apoptosis and is associated with a worse prognosis in hepatocellular carcinoma patients.

Since most endogenous circRNAs are not associated with ribosomes and the potency of cap-independent translation may be inefficient, further studies are needed to reveal the coding potential of circRNAs.

## 4. Function of circRNAs in Esophageal Cancer

Esophageal cancer is the sixth leading cause of cancer-related deaths worldwide. Squamous cell carcinoma is the most common histological type, particularly in Japan. Despite advances in multidisciplinary treatment, the prognosis of patients with ESCC remains poor. Thus, further studies are needed to find specific diagnostic biomarkers and arise novel therapeutic targets. Recently, several circRNAs have been reported to be differentially expressed between ESCC and adjacent normal tissues, indicating their key roles in ESCC development, progression, metastasis, and sensitivity to therapy.

### 4.1. Oncogenic circRNAs in Esophageal Cancer

Li et al. showed that ciRS-7 was significantly overexpressed in ESCC tissues [59]. The overexpression of ciRS-7 neutralized the suppressive effects of miR-7 on ESCC cell proliferation and migration both in vitro and in vivo. CiRS-7 sponges miR-7 to increase the expression of HOXB13 and then induces the phosphorylation of p65 in vitro. Overexpression of HOXB13 causes tumorigenesis in several cancer types, including prostate cancer [60]. The ciRS-7/miR-7/HOXB13/NF-κB/p65 pathway is aberrantly activated and induces malignant progression in ESCC [59]. CiRS-7 also increases the expression of KLF4, a transcription factor and a well-identified target of miR-7, and promotes the invasion of ESCC cells [61]. Sang et al. showed that ciRS-7 acts as a sponge for miR-876-5p and accelerates ESCC progression to enhance tumor antigen MAGE-A family expression. The online prediction database and luciferase reporter assay revealed that miR-876-5p directly targets *MAGE-A* family transcripts. Some of *MAGE-A* family genes have been shown to function as oncogenes in many cancer types, including ESCC [62,63]. CiRS-7 acts as cancer-promoting circRNA to inhibit the binding of miR-876-5p on its downstream target tumor antigen *MAGE-A* family in ESCC. MiR-1299 has also been reported to be a target of ciRS-7. Meng et al. [64] showed that overexpression of ciRS-7 markedly inhibits starvation- and rapamycin-induced autophagy in ESCC cells. Further research reveals that ciRS-7 act as a sponge for miR-1299 and then activates its downstream AKT/mammalian target of the rapamycin (mTOR) signaling pathway via modulating epidermal growth factor receptor (EGFR) expression. 

circCNOT6L (hsa_circ_0006168) consists of exons 2, 3, and 4 derived from *CCR4-NOT transcription complex subunit 6 like* (*CNOT6L*) on chromosome 4. Shi et al. showed that circCNOT6L is highly expressed in both ESCC tissues and cell lines [65]. High expression levels of circCNOT6L are associated with lymph node metastasis and tumor node metastasis (TNM) stage progression in patients with ESCC. circCNOT6L is mainly distributed in the cytoplasm and promotes ESCC proliferation, migration, and invasion through the miR-100/mTOR axis. Further studies showed that circCNOT6L targets miR-384 [66]. Knockdown of circCNOT6L enhanced miR-384 expression and inhibited tumor growth by inhibiting fibronectin 1 both in vitro and in vivo. 

CircPDE3B (hsa_circ_0000277) comprises exons 2–4 derived from *phosphodiesterase 3B* (*PDE3B*). Zhou et al. showed significantly increased circPDE3B expression in ESCC tissues and cell lines [67]. High circPDE3B expression correlates with advanced TNM stage and poor prognosis in ESCC patients. Knockdown of circPDE3B suppresses proliferation, invasion, and epithelial–mesenchymal transition (EMT) phenotypes of ESCC cells in vitro and in vivo. CircPDE3B acts as a miR-4766-5p sponge and then inhibits the binding of miR-4766-5p on its target gene, *laminin α1* (*LAMA1*). LAMA1 is frequently upregulated in multiple cancer types, such as colorectal carcinoma [68], melanoma [69], gastric cancer [70], and ESCC [71], and plays an important role in metastasis. Knockdown of circPDE3B decreases the expression levels of EMT-inducing factors, including MMP2/7/9, N-cadherin, Snail, Slug, and vimentin, and increases the expression of E-cadherin protein. Further studies have shown that circPDE3B also acts as a sponge for miR-873-5p, which targets *Sry-related high-mobility group box 4* (*SOX4*) mRNA [72]. SOX4 is known to regulate the Wnt/β-catenin pathway in many cancers [73]. CircPDE3B knockdown inhibits the Wnt/β-catenin pathway by regulating the miR-873-5p/SOX4 axis and suppresses cell progression and cisplatin resistance in ESCC cells in vitro and in vivo.

### 4.2. Tumor Suppressive circRNAs in EC

CircFAT1 (hsa_circ_0001461) is formed by the back-splicing of exon 2 of the *FAT1* gene and head-to-tail binding. Increasing evidence has shown that circFAT1 exerts tumor-suppressive effects in gastric cancer and colorectal cancer [74,75], and tumor-promoting effects in osteosarcoma, papillary thyroid cancer, and hepatocellular carcinoma [76,77,78,79]. Takaki et al. [80] showed that circFAT1 expression levels are significantly lower in ESCC tissues than in non-tumorous tissues. In prognostic analysis, high circFAT1 expression is an independent factor for better recurrence-free survival and cancer-specific survival. CircFAT1 acts as a tumor suppressor by sponging miR-548g to suppress ESCC cell invasion and migration in vitro. 

CircFAM120B (hsa_circ_0001666) is 2038 nucleotides long and originates from exons 2–4 of the *FAM120B* gene. CircFAM120B is downregulated in both ESCC tissues and plasma from patients with ESCC [81]. Expression of circFAM120B is negatively related to tumor size, and patients with low circFAM120B expression have a poor prognosis. Exogenous circFAM120B inhibits the proliferation, invasion, and migration of ESCC cells. CircFAM120B acts as a sponge for miR-661, which targets the tumor-suppressor gene *PPM1L*. CircFAM120B also binds to protein kinase R (PKR) and promotes its polyubiquitination and degradation. CircFAM120B negatively regulates the phosphorylation of p38 and the expression levels of N-cadherin and vimentin protein, whereas it positively regulates the protein level of E-cadherin. These results indicate that circFAM120B is a tumor-suppressive circRNA altering the miR-661/PPM1L axis and PKR/p38 MAPK/EMT pathway.

Meng et al. [82] found that ZEB1 inhibited the biogenesis of circDOCK5 (hsa_circ_0007618) by directly binding to the *DOCK5* and *EIF4A3* promoter. The expression levels of circDOCK5 are downregulated in ESCC tissues, and patients with low circDOCK5 expression levels have significantly shorter overall survival. CircDOCK5 increases the stability of miR-627-3p by functioning as a “reservoir”, and partially inhibits TGFB2 expression and TGF-β secretion, which further results in suppressed ZEB1 expression and EMT. These results indicate that ZEB1 downregulates circ-DOCK5 to facilitate metastasis by forming a positive feedback loop with TGF-β via miR-627-3p/TGFB2 signaling.

Since these studies were limited to a small number of cases, it is not yet clear whether these circRNAs act as oncogenes or tumor suppressors in ESCC. For example, Fan et al. [83] reported the tumor-suppressive function of ciRS-7, while ciRS-7 has been considered a tumor-promoting circRNA, as described above. They showed that the expression of ciRS-7 was significantly lower in ESCC tissues than in non-tumor tissues. Multivariate Cox regression analysis revealed that low ciRS-7 expression was associated with significantly poorer prognosis in patients with ESCC. Further studies are needed to clarify the function and prognostic impact of each circRNA on ESCC.

### 4.3. circRNAs as Diagnostic and Prognostic Biomarkers for Esophageal Cancer

Because of the high mortality rate associated with ESCC, early diagnosis and prognostic indicators are needed. Since circRNAs are abundantly expressed in cancer cells, they have the potential to be used as biomarkers for ESCC. Moreover, circRNAs are stable in human body fluids, including serum, plasma, urine, and microvesicles, indicating that they can be easily and repeatedly detected using noninvasive methods such as liquid biopsy [84]. Some studies have demonstrated circRNAs as potential biomarkers for the early diagnosis of ESCC and prediction of metastasis using clinical samples, including tissue and plasma from patients with ESCC (Table 1).

CircFNDC3B (hsa_circ_0006948), derived from exons 2, 3, and 4 of *fibronectin type III domain-containing protein 3B* (*FNDC3B*), is upregulated in ESCC tissues and cell lines [90]. Fluorescence in situ hybridization showed that circFNDC3B was preferentially expressed in the cytoplasm. High circFNDC3B expression was positively related to lymph node metastasis but not to other clinicopathological features such as age, T stage, sex, blood type, tumor location, and differentiation. Kaplan–Meier survival curves have shown that patients with high circFNDC3B expression had a significantly shorter overall survival [90]. The detection of circFNDC3B in ESCC tissues exhibited relatively good diagnostic performance with an area under the ROC curve (AUC), sensitivity, and specificity in EC tissues of 0.85, 0.74%, and 0.88%, respectively [90].

CircGSK3β (hsa_circ_0007986, also named circRNA_103443) is a back-splice variant of exons 3, 4, and 5 of the *Glycogen synthase kinase-3β* (*GSK3β*) gene located at 3q22.1. GSK3β is an important mediator of the Wnt signaling pathway and leads to the development and progression of many cancer types [109]. The expression of circGSK3β is frequently higher in ESCC tissues than in adjacent normal tissues [92]. Patients with high circGSK3β expression levels showed a worse TNM stage and lymph node metastasis. Kaplan–Meier survival curves have shown that patients with high circGSK3β expression had a significantly shorter metastasis-free survival (hazard ratio = 2.93) and overall survival (hazard ratio = 6.04). Compared to those in the healthy control group and with benign lesions, the abundance of circGSK3β in the plasma was significantly increased in patients with ESCC. The plasma levels of circGSK3β had relatively good diagnostic performance in ESCC tissues, with AUC, sensitivity, and specificity values of 0.78, 0.86%, and 0.58%, respectively. When plasma levels of circGSK3β were combined with a commonly used diagnostic marker, carcinoembryonic antigen (CEA), the AUC and specificity increased to 0.800 and 67.4%, respectively [92]. Moreover, the combination of circGSK3β with CEA markedly increased the sensitivity to 87.5% in the early stages of ESCC. Patients with recurrence/metastasis 10 months after surgery showed higher expression levels of circGSK3β compared to patients without recurrence/metastasis, indicating that quantification of plasma circGSK3β level is valuable to predict recurrence/metastasis of ESCC [92].

circIMMP2L (hsa_circ_0081964) is derived from exons 2–4 of the *Inner Mitochondrial Membrane Peptidase Subunit 2* (*IMMP2L*) gene. circIMMP2L has significantly higher expression in ESCC tissues from patients with an advanced TNM stage and lymph node metastasis [44]. The expression of plasma circIMMP2L in patients with advanced-stage ESCC or lymph node metastasis was significantly upregulated compared to that in healthy individuals. Quantification of circIMMP2L in plasma exhibited a relatively good diagnostic performance for the detection of lymph node metastasis, with an AUC of 0.865 [44].

Fan et al. [83] profiled circRNAs both in ESCC and non-tumor tissues and determined six circRNAs (hsa_circ_0062459, hsa_circ_0076535, hsa_circ_0072215, hsa_circ_0042261, hsa_circ_0001946, and hsa_circ_0043603) as the differentially expressed circRNAs. Among them, hsa_circ_0062459, hsa_circ_0001946, and hsa_circ_0043603 were detectable in the plasma. Furthermore, hsa_circ_0001946 and hsa_circ_0043603 could be useful for diagnostic biomarkers. The AUC, sensitivity, and specificity of hsa_circ_0001946 were 0.894, 92%, and 80%, respectively, while those of hsa_circ_0043603 were 0.836, 64%, and 92%, respectively. Hsa_circ_0001946 also predicted recurrence, overall survival, and disease-free survival in frozen and formalin-fixed, paraffin-embedded tissues.

A model based on the expression values of several circRNAs can improve the diagnostic and prognostic accuracy of ESCC. Wang et al. [87] established a prognostic risk model composed of circUSP13 (hsa_circ_0007541), circCDK11A (hsa_circ_0000005), circATG5 (hsa_circ_0077536), and circATXN10 (has_circ_0008199). The risk score of the selected circRNA signature demonstrated good performance in survival prediction and could be used as a sensitive prognostic biomarker in patients with ESCC, with an AUC of 0.839 [87].

Although the potential of circRNAs as ESCC biomarkers is becoming clear, their true value requires further validation in a larger cohort of clinical samples. Furthermore, in the absence of suitable biomarkers with high specificity and sensitivity at the same time, new diagnostic models must be established.

### 4.4. circRNAs as Potential Therapeutic Targets for Esophageal Cancer

The regulatory role of circRNAs in cancer is gradually becoming clearer, and circRNAs can be developed as effective therapeutic targets in the future. 

Several circRNAs have been reported as predictors of the response to radiation therapy and chemotherapy (Table 2). Hsa_circ_0014879, derived from *DDB1 and CUL4 associated factor 8* (*DCAF8*), is highly expressed in the radioresistant ESCC cell line KYSE-150R [110]. Hsa_circ_0014879 is involved in ESCC radioresistance via the miR-217/Wnt3 [110] and miR-519-3p/CDC25A pathways [111]. In contrast, hsa_circ_0000518 derived from *ribonuclease P RNA component H1* (*RPPH1*), also called circRNA_000167, is downregulated in KYSE-150R cells, suggesting that hsa_circ_0000518 enhances sensitivity to radiation therapy [112]. Cisplatin, fluorouracil, and paclitaxel are clinically used for the treatment of ESCC. CircDOCK1 (hsa_circ_0007142), derived from *Dedicator of cytokinesis 1* (*DOCK1*), is upregulated in cisplatin-resistant ESCC tissues and cells [113]. Upregulated circDOCK1 is involved in cisplatin resistance by upregulating LIM and SH3 protein 1 (LASP1) by miR-494-3p sponging. circCNOT6L is upregulated in paclitaxel-resistant ESCC cells [114]. circCNOT6L promotes paclitaxel resistance in ESCC by regulating the miR-194-5p/jumonji domain containing 1C (JMJD1C) axis, suggesting that circCNOT6L may be a therapeutic target for ESCC chemotherapy.

Rapidly increasing evidence has revealed that exosomes from resistant cells can render chemosensitive cells resistant to drugs. The structure of exosomes blocks RNA degradation and guarantees an efficient concentration of circRNAs. The size and membrane structure of exosomes also promote the absorption and fusion of cancer cells. Recently, Zang et al. [117] reported that exosome-mediated circPPFIA1 could induce chemoresistance in ESCC cells. CircPPFIA1 (has_circ_0000337), derived from *PTPRF interacting protein alpha 1* (*PPFIA1*), promotes cisplatin resistance in ESCC cells by modulating the miR-377-3p/JAK2 axis. CircPPFIA1-containing exosomes secreted from cisplatin-resistant esophageal cancer cells can promote chemoresistance, cell growth, and metastasis in cisplatin-sensitive ESCC cells both in vitro and in vivo.

Although specific therapeutic developments directly targeting circRNAs have not yet been reported, the use of synthetic circRNA sponges may be an effective strategy. Synthetic circRNAs containing miR-21 binding sites have been shown to cause a loss of miRNA function in vitro, suggesting that synthetic circRNA sponges have potential for the treatment of cancer patients [118].

## 5. Conclusions

To date, dozens of studies have determined that circRNAs are a large class of non-coding RNAs with a cell type-specific and tissue-specific expression that act as crucial regulators for gene expression. However, little is known regarding the functions and molecular mechanisms involved in cancer development and metastasis. The most commonly used markers are not specific or sensitive enough to be used as early diagnosis and prognostic indicators. The characteristics of circRNAs, such as their stability, abundance, and widespread distribution, make them a promising biomarker for the clinical diagnosis and prognosis of cancer. As reported for various miRNAs, individual circRNAs are unlikely to be sensitive and specific to all types of cancers, and one specific circRNA may not be sufficient for cancer diagnosis or prognosis prediction. Therefore, a combination of cancer-related circRNA panels could be useful as biomarkers for future clinical diagnoses. circRNAs also have great potential as therapeutic targets. A new and effective therapeutic approach may be to modulate the cellular function of endogenous circRNAs or to introduce synthetic circRNAs into cancer cells. Although specific therapeutic developments directly targeting circRNAs have not yet been established, further research will serve as a basis for clinical use.

## Figures and Tables

**Table 1 biomedicines-10-01643-t001:** Circular RNAs (circRNAs) as diagnostic and prognostic biomarkers for ESCC.

circRNA	CircBase ID	Sample	Dysregulation in ESCC	Clinicopathological Association	Potential Function	AUC	Sensitivity/ Specificity (%)	References
circAAAS	hsa_circ_0026611	69 serums	up	LNM ^1^	Prognosis/ Diagnosis	0.724	0.800/0.529	[85]
circALS2	hsa_circ_0001093	40 tissues	up	TNM stage, LNM ^1^, Tumor size	Prognosis			[86]
circATG5	hsa_circ_0077536	125 tissues	up		Prognosis			[87]
circCDK11A	hsa_circ_0000005	125 tissues	up		Prognosis			[87]
circUSP13	hsa_circ_0007541	125 tissues	up		Prognosis			[87]
circCCT3	hsa_circ_0014715	67 tissues	up	TNM stage, Differentiation, Vascular invasion	Diagnosis	0.722	0.49/0.91	[88]
circCDR1	hsa_circ_0001946	123 tissues	up	Age	Prognosis			[59]
circCHD2	has_circ_0000654	57 tissues	up	TNM stage, LNM ^1^	Diagnosis			[89]
circCNOT6L	hsa_circ_0006168	52 tissues	up	Tumor depth, LNM ^1^	Diagnosis			[65]
circCNOT6L	has_circ_0006168	30 tissues	up		Diagnosis			[66]
circFNDC3B	hsa_circ_0006948	153 tissues	up	LNM^1^	Prognosis/ Diagnosis	0.850	0.74/0.88	[90]
circGFPT1	has_circ_0120816	36 tissues	up	Tumor depth, LNM ^1^	Diagnosis			[91]
circGSK3β	hsa_circ_0007986	50 tissues	up	TNM stage, LNM ^1^	Prognosis/ Diagnosis	0.782	0.86/0.58	[92]
circHIPK3	hsa_circ_0000284	32 tissues	up	LNM ^1^, Tumor size, Differentiation	Diagnosis			[93]
circIMMP2L	hsa_circ_0081964	54 tissues, 54 plasmas	up		Prognosis/ Diagnosis	0.865		[44]
CircLPAR3	hsa_circ_0004390	10 tissues	up	TNM stage	Prognosis			[94]
circNOX4	has_circ_0023984	70 tissues	up	-	Prognosis			[95]
circNTRK2	hsa_circ_0087378	56 tissues	up	TNM stage, LNM ^1^	Prognosis/ Diagnosis			[96]
circOGDH	ha_circ_0003340	45 tissues	up	-	Diagnosis			[97]
circPDE3B	hsa_circ_0000277	92 tissues	up	TNM stage, LNM ^1^	Prognosis			[67]
circPDE3B	hsa_circ_0000277	58 tissues	up	TNM stage, LNM ^1^	Prognosis/ Diagnosis			[72]
circPPFIA1	hsa_circ_0000337	48 tissues	up	-	Diagnosis			[98]
circPRRX1	hsa_circ_0004370	25 tissues	up	Tumor size	Diagnosis			[99]
circRNF121	hsa_circ_0023404	74 tissues	up	TNM stage, Tumor depth, LNM ^1^, Vascular invasion	Prognosis			[100]
circSFMBT2	hsa_circ_0000211	39 tissues	up	TNM stage, LNM ^1^, Tumor size	Diagnosis			[101]
circSLC7A5	hsa_circ_0040796	87 tissues, 10 plasmas	up	TNM stage	Prognosis/ Diagnosis	0.772		[102]
circZDHHC5	hsa_circ_0004997	24 tissues, 20 plasmas	up	-	Diagnosis			[103]
circZFR	hsa_circ_0072088	83 tissues	up	TNM stage, Tumor depth, Tumor size, LNM ^1^	Diagnosis			[104]
circATXN10	hsa_circ_0008199	125 tissues	down		Prognosis			[87]
circBMI1	hsa_circ_0093335	10 tissues, 49 serums	down	LNM ^1^, Histological type	Diagnosis	0.726	0.96/0.47	[105]
circCDR1	hsa_circ_0001946	50 tissues, 50 plasmas	down	TNM stage, Tumor depth, Tumor size, LNM ^1^, Gender, CEA ^2^	Prognosis/ Diagnosis	0.894	0.92/0.80	[83]
circCNTNAP3	hsa_circ_0087104	60 tissues	down	-	Prognosis			[106]
circDOCK5	hsa_circ_0007618	100 tissues	down	-	Prognosis			[82]
circFAM120B	hsa_circ_0001666	130 tissues, 8 plasmas	down	Tumor size	Prognosis/ Diagnosis			[81]
circFAT1	has_circ_0001461	51 tissues	down	Age	Prognosis/ Diagnosis			[80]
circKRT19	hsa_circ_0043603	50 tissues, 50 plasmas	down	TNM stage, CEA ^2^	Diagnosis	0.836	0.64/0.92	[83]
circNTRK2	hsa_circ_0087378	50 tissues	down	TNM stage, LNM ^1^, Tumor size	Prognosis			[107]
circSMAD7	hsa_circ_0000848	36 tissues, 32 plasmas	down	TNM stage, LNM ^1^	Diagnosis	0.859	-	[108]

^1^ LNM: lymph node metastasis, ^2^ CEA: carcinoembryonic antigen.

**Table 2 biomedicines-10-01643-t002:** circRNAs as biomarker for radio- and chemo-sensitivity in ESCC.

circRNA	CircBase ID	Biological Function	Sample	References
**circRNAs as Radiosensitive or Radioresistance Biomarkers for EC**				
circDCAF8	hsa_circ_0014879	Radioresistance	KYSE150	[110]
circLIN52	hsa_circ_0000554	Radioresistance	KYSE150	[112]
circRPPH1	hsa_circ_0000518	Radiosensitive	KYSE150	[112]
**circRNAs as Chemoresistance Biomarkers for EC**				
circTMX4	hsa_circ_0001131	Cisplatin resistance	KYSE30, ECA109	[115]
circDOPEY2	hsa_circ_0008078	Cisplatin resistance	ESCC tissues, TE1, ECA109	[116]
circDOCK1	hsa_circ_0007142	Cisplatin resistance	TE1, KYSE410	[113]
circPDE3B	hsa_circ_0000277	Cisplatin resistance	ESCC tissues	[72]
circPPFIA1	has_circ_0000337	Cisplatin resistance	ESCC tissues, EC9706, KYSE30	[117]
circCNOT6L	has_circ_0006168	Paclitaxel resistance	KYSE150, ECA109	[114]

## Data Availability

Not applicable.

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
