# Peer review of "circRNA: A New Biomarker and Therapeutic Target for Esophageal Cancer"

_biomedicines, 2022, doi:10.3390/biomedicines10071643_

Round 1
Reviewer 1 Report
Shoda and colleagues demonstrated the potential of circRNAs as a biomarker for esophageal cancer. The manuscript is well-organized, well-written, and very informative to follow current updates of circRNAs in esophageal cancer. Authors described the biogenesis of circRNAs, their biological roles, and their functions in cancer development in detail, thereby providing valuable information to understand the significance of circRNAs in esophageal cancer.
Minor suggestions:
1. Please consider changing the title of section 3 (Biology of circRNAs --> biological roles of circRNAs).
2. Please revise the format of table 2. ('CircRNAs as chemoresistance biomarkers for EC' should be in bold.)
Author Response
Reviewer #1
Comment:
Shoda and colleagues demonstrated the potential of circRNAs as a biomarker for esophageal cancer. The manuscript is well-organized, well- written, and very informative to follow current updates of circRNAs in esophageal cancer. Authors described the biogenesis of circRNAs, their biological roles, and their functions in cancer development in detail, thereby providing valuable information to understand the significance of circRNAs in esophageal cancer.
Reply:
We thank the reviewer for his/her careful reading of our work and thoughtful comments. The following are point-by-point responses to the concerns raised.
Minor suggestion 1:
Please consider changing the title of section 3 (Biology of circRNAs --> biological roles of circRNAs).
Reply:
We wish to thank the reviewer for his/her helpful comment.
Following the suggestion of reviewer #1, we have changed the title of section 3 to "Biological roles of circRNAs.
Minor suggestion 2:
Please revise the format of table 2. ('CircRNAs as chemoresistance biomarkers for EC' should be in bold.)
Reply:
Following the suggestion of reviewer #1, we have fixed a mistake in table 2 (“CircRNAs as chemoresistance biomarkers for EC” is in bold). We are grateful to reviewer #1 for pointing out our mistake.
Reviewer 2 Report
The article ‘CircRNA: a new biomarker and therapeutic target for esophageal cancer’ discusses very important aspects related to circRNA. It should be stressed that writing is clear concise and interesting. The article has a correct composition. Authors present information about the biogenesis of circRNAs, their biological roles, and their functions in cancer development. A particularly important part of the manuscript is that which describes circRNA as a biomarker for esophageal cancer. The authors have prepared Tables 1 and 2 well by providing a large amount of information. Review is based mainly on the latest literature.Author Response
Reviewer #2
Comment:
The article ‘CircRNA: a new biomarker and therapeutic target for esophageal cancer’ discusses very important aspects related to circRNA. It should be stressed that writing is clear concise and interesting. The article has a correct composition. Authors present information about the biogenesis of circRNAs, their biological roles, and their functions in cancer development. A particularly important part of the manuscript is that which describes circRNA as a biomarker for esophageal cancer. The authors have prepared Tables 1 and 2 well by providing a large amount of information. Review is based mainly on the latest literature.
Reply:
We thank the reviewer for his/her careful reading of our work and thoughtful comments.